# Acceptance of COVID-19 Vaccination among Front-Line Health Care Workers: A Nationwide Survey of Emergency Medical Services Personnel from Germany

**DOI:** 10.3390/vaccines9050424

**Published:** 2021-04-23

**Authors:** André Nohl, Christian Afflerbach, Christian Lurz, Bastian Brune, Tobias Ohmann, Veronika Weichert, Sascha Zeiger, Marcel Dudda

**Affiliations:** 1Emergency Medical Services, Fire Brigade Oberhausen, 46047 Oberhausen, Germany; andre.nohl@bg-klinikum-duisburg.de (A.N.); christian.afflerbach@oberhausen.de (C.A.); christian.lurz@oberhausen.de (C.L.); 2Department of Emergency Medicine, BG Klinikum Duisburg, 47249 Duisburg, Germany; Sascha.Zeiger@bg-klinikum-duisburg.de; 3Helicopter Emergency Medical Service (HEMS), 47249 Duisburg, Germany; veronika.weichert@bg-klinikum-duisburg.de; 4Department of Anesthesiology and Critical Care, Evangelisches Krankenhaus Oberhausen, 46047 Oberhausen, Germany; 5Department of Trauma, Hand and Reconstructive Surgery, University Hospital Essen, 45147 Essen, Germany; bastian.brune@uk-essen.de; 6Emergency Medical Services, Fire Brigade Essen, 45139 Essen, Germany; 7Department of Research, BG Klinikum Duisburg, 47249 Duisburg, Germany; tobias.ohmann@bg-klinikum-duisburg.de; 8Department of Trauma Surgery, BG Klinikum Duisburg, 47249 Duisburg, Germany; 9Emergency Medical Services, Fire Brigade Duisburg, 47058 Duisburg, Germany

**Keywords:** EMS, COVID-19, Corona, vaccination, front-line, health-care workers, emergency medical services, hesitancy, SARS-CoV-2

## Abstract

Introduction: The aim of this study was to evaluate the COVID-19 vaccination acceptance of emergency medical services (EMS) personnel as front-line health care workers (HCW) in Germany. Several studies have shown low willingness for vaccination (e.g., seasonal influenza) among HCWs and EMS personnel. Methods: We created a web-based survey. The questions were closed and standardized. Demographic data were collected (age, sex, federal state, profession). Experience with own COVID-19 infection, or infection in personal environment (family, friends) as well as willingness to vaccinate was queried. Results: The sample includes *n* = 1296 participants. A willingness to be vaccinated exists in 57%, 27.6% participants were undecided. Our results show a higher propensity to vaccinate among the following groups: male gender, higher medical education level, older age, own burden caused by the pandemic (*p* < 0.001). Conclusions: Due to the low overall acceptance of vaccination by HCWs, we recommend that the groups with vaccination hesitancy, in particular, be recruited for vaccination through interventions such as continuing education and awareness campaigns.

## 1. Introduction

### 1.1. COVID-19 Pandemic

COVID-19, caused by severe acute respiratory syndrome coronavirus 2 (SARS-CoV-2), determines the totality of life worldwide [1]. Initially, the virus spread in Wuhan, China, but along the course of the viral disease, it was declared a worldwide pandemic by the World Health Organization (WHO) [2,3,4]. Severe acute respiratory syndrome coronavirus 2 is spread by direct contact or droplet transmission, attributable to respiratory particles. Airborne transmission can occur when respiratory aerosols are generated during patient respiratory activity or medical procedures. These aerosols contain particles that can travel greater distances and remain airborne for longer periods of time [5,6,7].

In spring 2020, the pandemic in Germany was narrowed down by strict regulations to not overload the health care system.

During summer, the situation in Germany and almost worldwide had become slightly less tense. Since incidence was increasing significantly through fall, the burden on the health care system also increased. A renewed lockdown with massive restrictions for the population was decided [8]. In several hotspots, there were temporary capacity shortages in hospitals [9].

Emergency medical services personnel have direct contact with all patient groups and is therefore classified as front-line health care workers [10]. In particular, with patients who have been infected by COVID-19 and require immediate medical attention. Due to contact with patients infected by COVID-19, the personnel themselves are at risk. It is known that increased numbers of health care workers (HCWs) have died from COVID-19 infection [11,12]. The staff themselves are a high risk to the patients they treat. Unrecognized COVID-19 infected personnel are especially endangering to seriously ill patients [13].

### 1.2. Emergency Medical Service in Germany

There are different specified qualifications in German emergency medical services (EMS): the emergency medical technician level (EMT-B: Rettungshelfer, EMT-I: Rettungssanitäter), the paramedic level (Rettungsassistent), the paramedic+ level (Notfallsanitäter), and the emergency physician (Notarzt). In the German EMS system, prehospital emergency care is primarily provided by paramedics, supported in life-threatening situations by an on-scene emergency physician. The competence of paramedics includes a range of advanced life support (ALS) treatments, which paramedics must perform until an emergency physician arrives on the scene. At that point, paramedics (and other medical personnel on scene) act under the direct medical supervision of the physician [14]. Other (non-life-threatening) emergencies are treated independently by the paramedics without a physician being on site.

### 1.3. Vaccination

Worldwide, several countries have made great efforts to advance vaccine development. The success of vaccination depends, on one hand, on a high level of acceptance in the population with a high number of people vaccinated and, on the other hand, on good immunization by the vaccine itself [15,16,17]. The first vaccine available in Germany was from Biontech/Pfizer: BNT162b2 [18]. At the time of our survey, it was initially known that the Biontech/Pfizer vaccine would be available.

MRNA vaccines are based on messenger ribonucleic acid (mRNA) and are a technology that stimulates the body’s immune response. These vaccines contain information from mRNA including the “blueprint” or code of a specific viral antigen. Based on the information, the body can produce this antigen itself: the mRNA transmits the information for the production of the antigen to our ribosomes in the cytoplasm, which manufactures proteins.

In the case of the mRNA vaccine from Biontech/Pfizer against COVID-19, the immune system recognizes the virus based on the coronavirus spike protein found on the surface of the virus. mRNA vaccines against COVID-19 are designed to provide our body with the code to produce a non-infectious viral spike protein. In doing so, they instruct the cellular machinery to stimulate a natural immune response. This immune response is achieved primarily by T-cells and the production of neutralizing antibodies, with the goal of preventing SARS-CoV-2 infection. When a vaccinated person later comes into contact with SARS-CoV-2, the immune system recognizes the surface structure and can fight and eliminate the virus. Neutralizing antibodies directed against SARS-CoV-2 bind to the virus and prevent the virus from entering the cell. T-cells help the immune system fight intracellular infections, and they can also kill the infected cells directly. Thus, unlike conventional vaccines, an mRNA vaccine does not contain any viral proteins itself, but only the information needed by our own cells to produce a viral trait that triggers the desired immune response. The mRNA technology has enabled the development of several candidate vaccines against COVID-19 [19,20,21,22,23].

The vaccination strategy in Germany is divided into several phases depending on the availability of a vaccine. In the first place, high-risk individuals with advanced age, or with correspondingly severe pre-existing conditions, should be vaccinated [24]. In second place are HCMs involved in the direct care of at-risk populations. This includes EMS personnel.

It is well known that in recent years, there has been an overall decrease in vaccination readiness in Europe [25]. A recent European survey on the willingness of the population to be vaccinated against COVID-19 shows disappointing results. In turn, concern about possible side effects increased [26,27]. Vaccination opponents spread misinformation, especially via social media. It is claimed that vaccination could negatively affect fertility in women. Other claims are that its infection is possible despite vaccination, an mRNA vaccine manipulates the human genome, a vaccination can trigger a tumor disease, private interests play a major role in the development of a vaccine, and many more [28,29,30,31,32].

Several studies have shown low willingness for seasonal influenza vaccination among HCWs and EMS personnel. Reasons often given are: self-determination, sufficient health status, fear of adverse effects, and concerns about safety and efficacy [33]. Inadequate vaccination compliance was also reflected among EMS personnel [34,35]. Therefore, we see it as very important to study vaccination acceptance among EMS personnel in the context of the global pandemic. To our best knowledge, COVID-19 vaccination acceptance among EMS personnel in Germany has not been studied.

The aim of this study was to evaluate the vaccination acceptance of emergency medical services personnel as front-line health care workers in Germany.

## 2. Materials and Methods

### 2.1. Questionnaire

A short questionnaire consisting of seven questions was created. All questions were closed and standardized. All questions had to be answered. The questionnaire could only be finished after complete answering. Demographic data were collected (age, sex, federal state, profession). Experience with own COVID-19 infection, or infection in personal environment (family, friends), burdened by the pandemic as well as acceptance to vaccinate was queried. Exposure to COVID-19 and acceptability for vaccination were queried using a 5-point Likert scale (1: strongly disagree, 2: somewhat disagree, 3: neither agree nor disagree, 4: somewhat agree, 5: strongly agree)

The questionnaire was created by consensus of the authors. This was followed by a trial run with three medical directors in emergency medical services and two paramedics. The comments of the participants were discussed and implemented for the final version.

Due to data protection, no personal data were stored. Participants were informed that answering the questions was voluntary and anonymous, and that the results would be scientifically evaluated and published. The survey was conducted online, and web based (Umfrageonline.com, enuvo GmbH, Zürich, Switzerland). The local ethics committee has approved the study.

### 2.2. Sample

Participation was open from 4 December 2020 to 15 January 2021. More than 250 medical directors in the EMS were informed about the survey and asked to post a QR code with a direct link to the survey in the EMS stations.

Professional groups are defined as (1) student/trainee, emergency medical technician–basic, (2) emergency medical technician–intermediate, (3) emergency medical technician–paramedic (nebulized medications, supraglottic airway management), (4) emergency medical technician–paramedic+ (i.v. medications, endotracheal intubation), and (5) emergency physician.

The various municipal EMS areas are medically managed by medical directors. The medical directors are particularly responsible for quality management. Therefore, they have contact with all the EMS employees who report to them. We contacted all available medical directors of the EMS. However, it is not possible to determine how many rescue stations and EMS personnel were reached. The total number of the population cannot be estimated with certainty because in the German EMS, especially in rural areas, volunteers are also used, who are not regularly active in the EMS. According to the Federal Employment Agency in Germany, there were over 73,333 EMS employees registered for social insurance in 2020 [36].

### 2.3. Statistics

This study employed univariate and bivariate analyses. The univariate analysis produced descriptive statistics, which were generated to produce summary tables for the study variables. Bivariate analysis was conducted as a cross tabulation between the dependent variables of interest. Multivariate logistic regression was performed to analyze the potential factors for vaccination willingness.

Statistical analysis was performed using IBM^®^ SPSS^®^ Statistics Version 27.0 (IBM Corporation, Armonk, NY, USA). A level of statistical significance of *p* <  0.05 was applied (Chi-square test).

### 2.4. Ethical Consideration

Participants were informed before answering the questions. They were informed that participation in the survey was voluntary and anonymous. Cancellation of participation was possible at any time. Furthermore, they were informed that the results would be evaluated and published.

## 3. Results

### 3.1. Demographic Data

The sample included *n* = 1296 participants. The demographic data are shown in Table 1. Most participants were men (*n* = 1013, 78.2%). In terms of professional groups, the largest group was emergency medical technician–paramedic+ level (*n* = 650, 50.2%). Age distribution showed that most of the participants were younger than 40 years (*n* = 827, 63.8%). Most participants were from North Rhine-Westphalia (*n* = 383, 29.6%), Lower Saxony (n = 229, 17.7%), Bavaria (*n* = 153, 11.8%), and Rhineland-Palatinate (*n* = 145, 11.2%). North Rhine-Westphalia, Bavaria, Baden-Württemberg, and Lower Saxony are the four most populous German states. Our results show a normal distribution in this area and is in line with official demographics in Germany. In comparison with other studies in the subject area of EMS, the age and gender were distributed equally [37,38].

### 3.2. Vaccination Willingness and Burden of COVID-19

A total of 51.5% of participants or close relatives had an infection themselves and 61% of participants felt burdened by the pandemic. A willingness to be vaccinated existed in 57%, while 27.6% participants were undecided (Table 2).

Men showed a higher willingness to vaccinate than women (60.7% vs. 44.2%, *p* < 0.001). The higher the age of the participants, the higher the willingness to be vaccinated (18–29 years: 54.5% vs. 60–69 years: 84.2%, *p* < 0.001). There were significant differences in vaccination willingness among different professional groups. The higher the level of medical education, the higher the willingness to vaccinate (emergency medical technician–basic: 42.9%; emergency medical technician–intermediate: 46.6%; emergency medical technician paramedic level: 54.8%; emergency medical technician paramedic+ level: 57.8%; emergency physician: 84.8%; *p* < 0.001). Those who felt burdened by the pandemic also indicated a higher willingness to be vaccinated (*n* = 447, 43.5%; somewhat agree and strongly agree) (Table 3). The potential factors influencing willingness to vaccinate (gender, age, professional group, own exposure to COVID-19) were calculated in a model using multivariate linear regression. The results are presented in Table 4.

## 4. Discussion

In our study, 57% participants showed vaccination acceptance. It should be noted that our survey was conducted at the time of the peak in COVID-19 cases in German hospitals from December 2020 to January 2021. Accordingly, there was also a peak in COVID-19 deployments in the German emergency medical services at this time. The strain on the emergency services was at its highest during this period. In particular, because the missions became significantly more strenuous due to additional protective measures and subsequent disinfection of emergency vehicles. There is a clear discrepancy between gender, professional groups and age. A vaccination rate of 60–70% is recommended for herd immunity [39]. According to this, the vaccination readiness in our sample is far too low.

Based on our results, we cannot prove the cause of vaccination hesitancy among emergency medical service personnel, but we can discuss it.

### 4.1. Differences in Vaccination Willingness

Our results show a higher acceptance to be vaccinated among the following groups:-males;-those with higher levels of education;-older individuals;-those who have first hand experience of the burden caused by the pandemic.

These results are comparable with other studies that have investigated the willingness to be vaccinated in the general population [40,41].

### 4.2. Vaccination Hesitancy among HCW

Several studies have shown that vaccination acceptance among HCWs has generally been low in the past, both dependent and independent of COVID-19 [40,42,43,44,45,46]. It is initially surprising that front-line HCWs, in particular, show low vaccination willingness, even though they are affected on a daily basis with patients suffering from COVID-19.

They are also exposed to the same external influences as the non-medical population. At the time of the survey, misinformation was increasingly circulating on the Internet and on social media. Medical personnel can also be susceptible to misinformation. It could, therefore, be explained that the willingness to vaccinate increases with the level of education. Personnel with a higher level of training seem to be better able to differentiate between information and misinformation due to their medical training. Higher levels of education lead to a better understanding as regards the efficiency and safety of vaccination. Thus, the willingness to vaccinate is highest among physicians. However, besides EMT-I and EMT-P, physicians were the smallest group in our sample. These results are concordant with other studies related to vaccination willingness among HCWs [47].

On the Internet, and especially on social media, vaccination opponents spread misinformation. This circumstance can generally lead to a reduction in the willingness to be vaccinated [28,29,48,49,50]. It has been widely disseminated that COVID-19 vaccination has effects on fertility in childbearing young women. The problem with spreading false information is that it is initially not refutable by lay people.

Especially in elderly patients, infection can lead to severe effects [3,51,52,53]. Therefore, this can explain the increase in willingness to be vaccinated with age. On the other hand, young people feel healthy, and therefore, exhibit increased vaccine hesitancy [54,55].

Pichon was able to show similar results in a study of HCWs in Italy. Again, the willingness of HCWs to be vaccinated increased with age and educational level [56]. However, other studies have shown that vaccination willingness in relation to age may also depend on the pathogen itself. For example, older HCWs appear to have a higher propensity to be vaccinated against influenza, but this may decrease with age for measles [57]. A French study showed a similar difference in vaccination willingness across pathogens [58]. The reasons for vaccination hesitancy in HCWs have been broadly studied. One of the most important concerns raised in other studies, is fear of side effects from the vaccines. New vaccines were rejected due to a perceived lack of testing for vaccine safety and efficacy. In addition, while health workers expressed great trust in health authorities, they also expressed a strong distrust of pharmaceutical companies due to perceived financial interests and a lack of communication about side effects [41,42,43,46,54,55,59,60,61].

However, other studies related to influenza vaccination have shown that those who are vaccinated perceive themselves as more knowledgeable about vaccination and show a willingness to protect both themselves and their contacts. They also consider vaccines to be effective and influenza to be a potentially serious threat [59,62,63].

Medical personnel are highly stressed by the pandemic. On the one hand, the workload is increasing, and on the other hand, colleagues are exposed to increased psychological pressure due to the severe courses of illness experienced by patients. Daily routines have been made more difficult by the continuous crisis. It is therefore not surprising that the majority of participants stated that they themselves were burdened by the pandemic. It is also understandable that the willingness to be vaccinated is statistically higher in this collective.

### 4.3. Addressing Vaccination Hesitancy

Strategies to address vaccination hesitancy consist of multiple components. Jarrett et al. showed in a review that dialogue-based strategies, in particular, and the promotion of knowledge and awareness can increase vaccination acceptance [64].

Success in increasing vaccination may result from a combination of the following interventions: (1) directly contacting non-vaccinated populations [65]; (2) strengthening knowledge and awareness of vaccinations [66]; (3) improving comfort and access to vaccinations [67]; (4) targeting specific populations (e.g., HCW) [68]; (5) mandated vaccinations or sanctions against non-vaccination [69]; (6) engaging religious or other influential leaders to promote vaccinations [70]

### 4.4. Limitations and Strengths

This is one of the very first studies—both performed in Germany and worldwide—that has evaluated the acceptance towards SARS-CoV-2 vaccination among emergency physicians and emergency medical service personnel. Our sample, compared to similar studies, is larger than average.

No data were collected on participants’ work experience. Work experience could influence vaccination willingness independent of education.

The conditions during the survey could not be controlled: Whether the participant was distracted; other people were present, influencing the processing; or whether some people participated more than once could not be traced. Participation required a computer or mobile device with internet access. Individuals could participate from different computers and, conversely, several people in a household can share the same computer.

## 5. Conclusions

The results of our survey showed that, in particular, women, young people, and less qualified medical staff could be associated with refusing to receive COVID-19 vaccination. Furthermore, our study showed that vaccination acceptance among medical personnel is not sufficient for herd immunity. Due to the low overall number of people willing to be vaccinated, we recommend that the above-mentioned groups, in particular, be recruited for vaccination through interventions such as continuing education and awareness campaigns.

With the results of our study, we can contribute to the fight against the global COVID-19 pandemic. Further studies are needed to investigate the lack of vaccination preparedness. Evaluation of interventions will be necessary to increase the effectiveness of these interventions.

## Figures and Tables

**Table 1 vaccines-09-00424-t001:** Demographic data.

	Number	Percent
Gender	*n*	%
Female	283	21.8
Male	1013	78.2
Professional group		
Students/trainees	81	6.3
EMT-B	28	2.2
EMT-I	315	24.3
EMT-P	104	8
EMT-P+	650	50.2
Emergency physician	118	9.1
Age distribution (years)		
18–29	389	30
30–39	438	33.8
40–49	294	22.7
50–59	156	12
60–69	19	1.5
State		
Baden-Wuerttemberg	80	6.2
Bavaria	153	11.8
Berlin	16	1.2
Brandenburg	29	2.2
Bremen	6	0.5
Hamburg	14	1.1
Hesse	71	5.5
Mecklenburg-Western Pomerania	23	1.8
Lower Saxony	229	17.7
North Rhine-Westphalia	383	29.6
Rhineland-Palatinate	145	11.2
Saarland	10	0.8
Saxony	34	2.6
Saxony-Anhalt	36	2.8
Schleswig-Holstein	40	3.1
Thuringia	25	1.9

*EMT-B* emergency medical technician–basic. *EMT-I* emergency medical technician–intermediate. *EMT-P* emergency medical technician–paramedic level (nebulized medication, supraglottic airway). *EMT-P+* emergency medical technician–paramedic level (intravenous medication, endotracheal intubation).

**Table 2 vaccines-09-00424-t002:** COVID-19.

COVID-19		
	Frequency	Percent
Participants were infected themselves or close relatives		
no	627	48.4
yes	668	51.5
Participants feel burdened by the pandemic		
Strongly disagree	36	2.8
Somewhat disagree	110	8.5
Neither agree nor disagree	358	27.6
Somewhat agree	401	30.9
Strongly agree	390	30.1
Vaccination acceptance		
Strongly disagree	193	14.9
Somewhat disagree	100	7.7
Neither agree nor disagree	262	20.2
Somewhat agree	246	19
Strongly agree	493	38

**Table 3 vaccines-09-00424-t003:** Cross-tabulation of vaccination willingness.

			Level of Agreement for Vaccination					*p*-Value
			Strongly Disagree	Somewhat Disagree	Neither Agree nor Disagree	Somewhat Agree	Strongly Agree	
Gender	Female	*n*	55	35	68	56	69	<0.001
		%	19.4	12.4	24	19.8	24.4	
	Male	*n*	138	65	194	190	424	
		%	13.6	6.4	19.2	18.8	41.9	
Age (years)	18–29	*n*	53	42	82	84	128	<0.001
		%	13.6	10.8	21.1	21.6	32.9	
	30–39	*n*	83	33	96	71	154	
		%	19	7.6	22	16.2	35.2	
	40–49	*n*	40	19	60	57	117	
		%	13.7	6.5	20.5	19.5	39.9	
	50–59	*n*	16	6	22	29	83	
		%	10.3	3.8	14.1	18.6	53.2	
	60–69	*n*	1	0	2	5	11	
		%	5.3	0	10.5	26.3	57.9	
Professional group	Student	*n*	8	10	15	24	24	<0.001
		%	9.9	12.3	18.5	29.6	29.6	
	EMT-B	*n*	4	3	9	4	8	
		%	14.3	10.7	32.1	14.3	28.6	
	EMT-I	*n*	61	31	76	47	100	
		%	19.4	9.8	24.1	14.9	31.7	
	EMT-P	*n*	14	7	26	12	45	
		%	13.5	6.7	25	11.5	43.3	
	EMT-P+	*n*	99	47	127	124	251	
		%	15.3	7.3	19.6	19.1	38.7	
	Emergency physician	*n*	7	2	9	35	65	
		%	5.9	1.7	7.6	29.7	55.1	
Participants feel burdened by the pandemic	Strongly disagree	*n*	16	1	8	1	10	<0.001
		%	44.4	2.8	22.2	2.8	27.8	
	Somewhat disagree	*n*	19	10	12	29	40	
		%	17.3	9.1	10.9	26.4	36.4	
	Neither agree nor disagree	*n*	42	26	78	74	138	
		%	11.7	7.3	21.8	20.7	38.5	
	Somewhat agree	*n*	42	35	86	91	147	
		%	10.5	8.7	21.4	22.7	36.7	
	Strongly agree	*n*	74	28	78	51	158	
		%	19	7.2	20.1	13.1	40.6	

*EMT-B* emergency medical technician–basic. *EMT-I* emergency medical technician–intermediate. *EMT-P* emergency medical technician–paramedic level (nebulized medication, supraglottic airway). *EMT-P+* emergency medical technician–paramedic level (intravenous medication, endotracheal intubation).

**Table 4 vaccines-09-00424-t004:** Multiple linear regression.

	RG	*p*-Value	95.0% CI	
			Lower Limit	Upper Limit
Gender	0.446	<0.001	0.257	0.636
Age	0.092	0.026	0.011	0.172
Professional Group	0.081	0.013	0.017	0.144
Own Burden	0.029	0.433	–0.044	0.102

CI: confidence interval, RG: regression coefficient.

## Data Availability

Data are available from andre.nohl@bg-klinikum-duisburg.de.

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
