# Peer review of "Acceptance of COVID-19 Vaccination among Front-Line Health Care Workers: A Nationwide Survey of Emergency Medical Services Personnel from Germany"

_vaccines, 2021, doi:10.3390/vaccines9050424_

Round 1
Reviewer 1 Report
Given the timing, the paper is of immediate interest and importance to German healthcare, German society, and the rest of the world. The policymakers and the healthcare sector will find these insights useful and need to address this hesitancy immediately. For that purpose, I support publishing the article.
Otherwise, the study has little to offer in the long-term and overall literature. The paper has weaknesses on the following fronts: a) it only reports findings of a short survey with seven questions with Likert scale analyzed with bivariate crosstabs.; b) the questions lack depth and don't reveal insights into new concepts; c) the paper does not go in-depth into how this hesitancy can be overcome.
I would recommend that the authors address my last point on proposing strategies to overcome hesitancy. They can cite examples of how attempts have been made in other vaccine situations in Germany or other countries.
Author Response
Dear reviewer,
Thank you very much for giving us the opportunity to revise and resubmit our manuscript. You made very helpful recommendations and we really have to thank you for helping us to improve our manuscript. We really appreciate this!
See our detailed point-by-point response below. Changes in the revised manuscript are marked-up in yellow.
c) the paper does not go in-depth into how this hesitancy can be overcome.
I would recommend that the authors address my last point on proposing strategies to overcome hesitancy. They can cite examples of how attempts have been made in other vaccine situations in Germany or other countries.
4.3 Addressing vaccination hesitancy
Strategies to address vaccination hesitancy are multi-component. Jarrett et al. showed in a review that dialogue-based strategies, in particular, and the promotion of knowledge and awareness can increase vaccination readiness (54).
Reviewer 2 Report
Materials and methods
- Page 3, line 122, how was the questionnaires developed, what were the questions asked and was the questionnaire piloted and validated?
- Page 3, line 123, working experience and previous vaccination behaviours of other vaccines are important factors associated with vaccine acceptance of HCWs, which could be one of the major limitations in the survey as there were no such data.
- Page 3, line 126, please describe the scale in English.
- Page 3, line 134, please provide the details of sampling method in this section.
- Page 3, line 143, apart from the cross-tabulation between vaccine acceptance and the independent variables, a multiple regression should be considered to determine the association of these variables and vaccine acceptance with adjustment of other potential confounders.
Results
- Page 4, line 157, please describe in the text or use a flow chart to show how the survey arrived at current sample size (n=1296). How many HCWs were approached? How many were excluded because they are not eligible participant or provided invalid/incomplete answers? Etc.
- Page 4, line 165, the authors have mentioned the demographical characteristics matched to demographics in Germany, does it mean demographics of general population? Could the authors also compare the age, gender, and/or professional group distribution of the sample with the statistics of HCWs or emergency room personnel in Germany? And if there any difference, does it lead to selection bias and any measures taken by the authors to reduce the bias?
- Page 4, line 167, table 1 demographic data, the abbreviations for these emergency medical technicians should be better cross-referenced in the preceding sections or explained in a foot-note of the Table.
- Page 5, line 174, is there any difference in vaccine acceptance across different emergency rooms, states or larger regions? And it could provide more information if the authors can comment on the relationship of this difference and the COVID-19 epidemic level of different areas.
Discussions
- Page 7, line 205, comparison of the results of this study and other studies on HCWs or emergency room personnel should also be made here.
- Page 7, line 212-220, how do the authors draw conclusions on the relationship among education level/professional group, misinformation, and vaccine acceptance, especially when there are no data on information sources or susceptibility to misinformation of the respondents?
- Page 7, line 221, following last comment, the authors has mentioned that “Women in particular were apparently susceptible to misinformation”, is there any reference to justify this statement? The authors also mentioned that “The significant difference is clear according to our statistical analysis and also described in other studies in relation to vaccination hesitancy”, but there are no variables regarding the influence of social media or misinformation in the survey data, and the authors did not cite any study on this matter.
- There are multiple statements in Discussion section lack of citations, which should be added.
- What the authors plan to do or suggest to improve the vaccination acceptance of emergency room personnel, based on the findings of this study?
Reviewer 3 Report
I was invited to revise the paper entitled "Acceptance of COVID-19 Vaccination among Front-Line Health-Care Workers: A Nationwide Survey of Emergency Medical Services Personnel from Germany ". It was a cross-sectional study aimed to investigate the willingness to accept covid19 vaccination among German HCW. This paper focus on an important topic for Public Health and it is the first paper conducted in Germany about this topic in my knowledge. These results can improve the knowledge in this field.
Despite this, the paper needs some improvement:
- The aim ofthe study should be reported at the end of Introduction section;
- Line 125 was reported in german language. Please translate it;
- Sample size estimation was missing;
- Validation of the questionnaire was needed. It was a Likert scale so Crombach's alpha should be calculated;
- In addition to the reported analysis, I suggest to summarize all answers of the questionnaire in "Strongly/Somewhat agree" vs others and perform a logistic regression to evaluate associated factors for each question;
- Statistical analysis section should be improved. Methods should be better described;
- About discussions, I suggest to comment about two missing important points: a) About gender differences in vaccine uptake among HCW (10.3390/vaccines8020248); b) about differences with other european country such as Austria (10.1080/21645515.2016.1168959) and France (10.1016/j.vaccine.2012.04.098)
Round 2
Reviewer 3 Report
The paper is now acceptable for publication